# Alkaline Dilution Alters Sperm Motility in Dairy Goat by Affecting sAC/cAMP/PKA Pathway Activity

**DOI:** 10.3390/ijms24021771

**Published:** 2023-01-16

**Authors:** Qifu He, Feng Gao, Shenghui Wu, Shaowen Wang, Zhiming Xu, Xuerui Xu, Tianyang Lan, Kang Zhang, Fusheng Quan

**Affiliations:** 1College of Veterinary Medicine, Northwest A&F University, Yangling 712100, China; 2Key Laboratory of Animal Biotechnology of the Ministry of Agriculture, Yangling 712100, China; 3Key Laboratory of Animal Biotechnology, Northwest A&F University, Taicheng Road, Yangling 712100, China

**Keywords:** sperm, pH, Ca^2+^, sAC/cAMP/PKA, dairy goat

## Abstract

In dairy goat farming, increasing the female kid rate is beneficial to milk production and is, therefore, economically beneficial to farms. Our previous study demonstrated that alkaline incubation enriched the concentration of X-chromosome-bearing sperm; however, the mechanism by which pH affects the motility of X-chromosome-bearing sperm remains unclear. In this study, we explored this mechanism by incubating dairy goat sperm in alkaline dilutions, examining the pattern of changes in sperm internal pH and Ca^2+^ concentrations and investigating the role of the sAC/cAMP/PKA pathway in influencing sperm motility. The results showed that adding a calcium channel inhibitor during incubation resulted in a concentration-dependent decrease in the proportion of spermatozoa with forward motility, and the sperm sAC protein activity was positively correlated with the calcium ion concentration (r = 0.9972). The total motility activity, proportion of forward motility, and proportion of X-chromosome-bearing sperm decreased (*p* < 0.05) when cAMP/PKA protease activity was inhibited. Meanwhile, the enrichment of X-chromosome-bearing sperm by pH did not affect the sperm capacitation state. These results indicate that alkaline dilution incubation reduces Ca^2+^ entry into X-sperm and the motility was slowed down through the sAC/cAMP/PKA signaling pathway, providing a theoretical foundation for further optimization of the sex control method.

## 1. Introduction

Goat’s milk is similar to human milk in terms of nutritional content and ease of digestion and is often the first choice for consumers [1,2]. In dairy goat breeding, male kids increase the feeding cost of the farm, and their value is lower than that of female kids; therefore, breeding more female kids can improve the economic benefits of a farm [3,4]. Previous studies have shown that the environmental pH can affect the motility and metabolism of rabbit sperm [5], hamster sperm [6], and human sperm [7,8]. X-chromosome-bearing sperm and Y-chromosome-bearing sperm exhibit different motility activities under different pH conditions [9,10], affecting the sex ratio of offspring. In our previous study, we found that dairy goat sperm incubated in alkaline diluent was enriched in X-chromosome-bearing sperm and that the quality and performance of the enriched X-chromosome-bearing sperm were not affected by the alkaline pH; consequently, the number of female embryos and female kids was significantly increased [11].

For the most part, homeostasis is accomplished by the interplay of multiple transporters that either extrude or import proton equivalents and are attuned to the physiological state of the cell by coupling either physically or functionally to metabolic enzymes [12]. Sperm motility is affected by internal pH (pHi) homeostasis, and its regulation mechanism is complex. Membrane proteins that mediate the transmembrane transport of H^+^ and HCO_3_^−^ ions have been reported in mammalian sperm [13,14], and the pHi of sperm directly affects the [Ca^2+^]i concentration [15,16]. Ca^2+^ plays the role of second messenger in sperm flagella swing, swimming speed, sperm capacitation, and the acrosome reaction. [Ca^2+^]i at concentrations of 10–40 nM triggers symmetrical beating of flagella and induces hyperactivation at 100–300 nM [12,17]. Sperm motility is mainly regulated through the activation of soluble adenylate cyclase sAC (resulting in cAMP synthesis and subsequent PKA activation), increased tyrosine phosphorylation, maintenance of sperm mitochondrial function, and ATP production [18,19,20,21]. However, changes in cytoplasmic pH, calcium ion concentration, and the mechanism by which the X/Y-chromosome-bearing sperm motility activity of dairy goat spermatozoa is affected by alkaline conditions remain unclear.

This study was designed to detect the changes in the internal pH and calcium ion concentration of dairy goat sperm during incubation in alkaline diluent. Calcium ion channel inhibitors and sAC/cAMP/PKA pathway protease inhibitors were utilized to detect sperm motility activity and the proportion of X-chromosome-bearing sperm in the upper and lower layers of the test tube. The elucidation of the mechanism of X-chromosome-bearing sperm enrichment under alkaline conditions will create a theoretical foundation for further optimization of the sex control method.

## 2. Results

### 2.1. Changes in pHi, Motility, and Proportion of X-Chromosome-Bearing Sperm during Incubation at pH 7.4

We established in a previous study [11] (Figure 1A) that the pHi of the sperm in the upper and lower layers of a test tube increased to 6.9 (initial 5.01) over 10 min and then remained unchanged when incubated in a diluent of pH 7.4 (Figure 1B,C). The total motility and forward motility of spermatozoa in the upper layer of the tube were found to be significantly higher than that of spermatozoa in the lower layer (Figure 1D,E). The proportion of X-chromosome-bearing sperm increased with the incubation time and reached a peak value of 70.00% ± 2.70% at 40 min (Figure 1F). (Detailed data in Appendix A)

### 2.2. Changes in [Ca^2+^]i Concentration during Sperm Incubation

Flow cytometry was used to detect the changes in the calcium ion concentrations after incubation of sperm at different pHs. It was found that when sperm was incubated at pH 6.8 and pH 7.4 for 40 min, the proportion of sperm with high calcium ion concentrations increased. At pH 7.4, the increase in the upper layer of sperm was significantly higher than that in the lower layer (Figure 2A–D).

The OCPC colorimetric method was also used to detect the [Ca^2+^]i concentration in the sperm, and the standard curve was established as y = 2.475x + 0.03945 (R^2^ = 0.9924) (Figure 3A). These results were consistent with those from flow cytometry (Figure 3B1,B2). When NNC was added during incubation at pH 7.4, it was found that the calcium ion concentration and the proportion of forwardly motile sperm in the lower layer of spermatozoa decreased with an increase in the NNC concentration (r = 0.9943), and the proportion of X-chromosome-bearing sperm also decreased with an increase in the NNC concentration (Figure 3C,D). (Detailed data in Appendix A)

### 2.3. Correlation between sAC Protein Activity and [Ca^2+^]i Concentration in Sperm

Using the genomic DNA and the protein from the frozen semen of Holstein dairy cows as positive controls, the PCR amplification and Western immunoblotting experiments showed that the sAC gene and sAC protein existed in the sperm of dairy goats (Figure 4A,B). After incubating sperm at pH 7.4 with a prolonged incubation time, the activity of the sAC protein in the upper and lower layers of sperm increased, and the increase in the upper layer was significantly higher than that in the lower layer (Figure 4C). The correlations between the activity of the sAC protein in the upper and lower layers of the spermatozoa and the internal pH were r = 0.7667 and r = 0.7416, respectively (Figure 4D,E), and the correlations with the internal calcium ion concentration were r = 0.9831 and r = 0.9972, respectively (Figure 4F,G). (Detailed data in Appendix A)

### 2.4. Correlation between Sperm sAC Activity and cAMP and PKA Protease Activities 

According to the instructions of the cAMP content detection kit and the PKA activity detection kit, the standard curve of cAMP content and OD533nmRFU was established using the following standard: y = −3.1477x^4^ + 88.341x^3^ − 207.08x^2^ − 1150.8x + 3137.9, R^2^ = 0.9912 (Figure 5A). A standard curve of PKA concentration versus OD450nmRFU was established as follows: y = 0.0543x + 0.1771, R^2^ = 0.9936 (Figure 5B). When incubated at pH 7.4, the correlation coefficients between the sAC activity of the upper and lower layers of spermatozoa and the cAMP content of the corresponding layer were r = 0.9780 and r = 0.9938, respectively, and those between the sAC activity of the upper and lower layers of the spermatozoa and the corresponding PKA protease activity were r = 0.9869 and r = 0.9979, respectively (Figure 5C1,C2,D1,D2). 

The pPKAs and pY contents of the sperm incubated at pH 7.4 were significantly higher than those incubated at pH 6.8 (Figure 6A1,A2,B1,B2). When incubated at pH 7.4, the correlation coefficients between the sAC activity of the upper and lower layers of spermatozoa and the proportion of forward motility were r = 0.9935 and r = 0.9964, respectively, and those between the sAC activity of the upper and lower layers of the spermatozoa and the proportion of X-chromosome-bearing sperm were r = −0.9892 and r = 0.9972, respectively (Figure 6C,D). (Detailed data in Appendix A)

### 2.5. Sperm Motility following the Inhibition of cAMP and PKA Protease Activity

SQ22536 was added to the pH 7.4 dilution solution to inhibit the cAMP content of sperm, and samples were incubated for 40 min. The sperm cAMP content in the upper and lower layers of the test tube in the inhibitor group was significantly decreased compared with that in the no inhibitor group (*p* < 0.01) (Figure 7A). When incubated at pH 7.4, the correlation coefficients between the cAMP content of the upper and lower layers of the spermatozoa and the PKA activity of the corresponding layer were r = 0.9672 and r = 0.9338, respectively (Figure 7B1,B2). When incubated at pH 7.4, the correlation coefficients between the cAMP content of the upper and lower layers of the spermatozoa and the proportion of forward motility were r = 0.9707 and r = 0.9525, respectively, and those between the cAMP content of the upper and lower layers of the spermatozoa and the proportion of X-chromosome-bearing sperm were r = −0.9800 and r = 0.9217, respectively (Figure 7C1,C2,D1,D2). When sperm were incubated at pH 7.4, the contents of p-PKAs and pY in the upper sperm were significantly higher than those in the lower sperm (*p* < 0.01) (Figure 7E,F).

H89 was added to the pH 7.4 diluent to inhibit PKA protease activity, and samples were incubated for 40 min. The sperm PKA protease activity in the upper and lower layers of the test tube in the inhibitor group was significantly decreased compared with that in the no inhibitor group (*p* < 0.01) (Figure 8A). When incubated at pH 7.4, the correlation coefficients between the PKA protease activity of the upper and lower layers of the spermatozoa and the proportion of forward motility were r = 0.9976 and r = 0.9981, respectively, and those between the PKA protease activity of the upper and lower layers of spermatozoa and the proportion of X-chromosome-bearing sperm were r = −0.9913 and r = 0.9907, respectively (Figure 8B1,B2,C1,C2). When sperm were incubated at pH 7.4, the contents of p-PKAs and pY in the upper sperm were significantly higher than those in the lower sperm (*p* < 0.01) (Figure 8D,E). (Detailed data in Appendix A)

### 2.6. Enriched Sperm Capacitation Status

CTC staining was used to test the capacitation status of the spermatozoa. The proportions of intact and non-capacitated spermatozoa in the upper and lower parts of the tube after 40-min incubation at pH 7.4 were 85.05% ± 2.25% and 90.25% ± 2.75%, respectively, which showed no significant difference from the control group (*P >* 0.05) (Figure 9B). (Detailed data in Appendix A)

## 3. Discussion

In dairy goat breeding, to obtain greater economic benefits, it is necessary to breed more female kids than males. The most direct method to achieve this is to separate the X- and Y-chromosome-bearing sperm prior to insemination [3,4]. Flow cytometry is currently the most effective and widely used sperm separation method [22,23]. The chromosomal DNA content of goat X-chromosome-bearing sperm is 4.4% ± 0.03% higher than that of goat Y-chromosome-bearing sperm [24] and is therefore, in principle, easy to separate by flow cytometry. However, the sorting speed is slow, and it cannot meet the insemination requirements of tens of millions of insemination doses per insemination. The equipment is expensive and requires professional operation, so it cannot be widely used in the dairy goat industry [25]. 

Previous studies have reported that artificial insemination after incubation of rabbit sperm at different pHs significantly changes the sex ratio of rabbit offspring [5]. Another study reported that, when the pH value of the reproductive tract of hamsters was increased, the proportion of male offspring decreased, showing a significant negative correlation [6]. The ratio of X/Y-chromosome-bearing sperm has also been reported to change after incubation of human sperm at different pHs [7,8]. A previous study in our laboratory revealed that, when dairy goat sperm were incubated in alkaline diluent, X-chromosome-bearing sperm could be effectively enriched in the lower layer of the test tube [11]. However, we did not study the specific mechanism in this previous study.

The motility and fertilization potential of mammalian sperm are regulated by ion homeostasis, including sperm pHi, membrane voltage, and [Ca^2+^]i [8,12], which are controlled by ion channels and transporters. Sperm pHi is directly related to motility, and previous studies have reported higher values for sperm kinetic parameters, membrane integrity, and mitochondrial activity at pH 7 and 7.5, whereas motility and other parameters have been found to be significantly lower at pHs below 6.5 or above 8 [26,27,28]. The activity of ion channels (Ca^2+^, k^+^, Na^+^, etc.) on the sperm membrane surface [12,29] and the phosphorylation level of the VSP protein in the sperm tail can differ widely [30,31]. Intracellular alkalinization is critical not only for sperm hyperactivation and acrosome reactions but also for basal sperm motility, as the ability of dynein to hydrolyze ATP and provide axial filament bending increases significantly with increasing pHi [32,33]. These pH-regulating proteins can be classified into two categories: (1) Membrane proteins that mediate H^+^ transmembrane transport, including proton channels (Hv1), proton pumps, and Na^+^/H^+^ exchangers (NHE); (2) Membrane proteins that mediate HCO_3_^−^ ion transmembrane transport, including SLC4 family proteins, SLC26 family proteins, and cystic fibrosis transmembrane transduction regulators (CFTR) [34]. NHE have been detected in the sperm of sheep [35], mice [36,37], and pigs [38]. Under physiological conditions, NHE-mediated cytoplasmic alkalinization should enhance the pH-sensitive ion channels, KSper and CatSper, thereby regulating sperm function [36,39]. Hv1 is present in human sperm [40,41] and bull sperm [13], and it is essential for the alkalization of sperm flagella and further activation of sperm motility through a Ca^2+^-dependent pathway. CFTR ion channels exist in human spermatozoa [19,41] and mouse spermatozoa [42], and are involved in changes in H^+^/HCO_3_^−^ in sperm [14]. In this study, it was found that when dairy goat sperm was incubated at pH 7.4, the internal pH of the spermatozoa increased to 6.9 (initially 5.01) within 10 min and then remained unchanged. The proportions of total motility and forward motility of the upper layer spermatozoa were significantly higher than those of the lower layer spermatozoa; therefore, this phenomenon was not caused by the change in pHi of the sperm alone.

In spermatozoa, swimming behavior is controlled by rises in Ca^2+^, which change the flagellar beat pattern via Ca^2+^-sensing protein calaxins [43]. Ca^2+^ is required for sAC/cAMP/PKA pathway activation [44], and Ca^2+^ binds to the calmodulin present in the sperm head and flagellum, thus activating other additional phosphorylation cascades regulated by calmodulin kinase as well as the maintenance of sperm mitochondrial functioning and ATP production [45], which are critical for sperm cell motility. Calcium influx is mainly driven by sperm-specific pH-sensitive voltage-gated Ca^2+^ channels (CatSper), and a small proportion of calcium influx comes from the release of calcium ions in the sperm [12]. Changing the pHi from acidic to basic within the physiological range (6.0 to 7.4) has been found to induce a seven-fold increase in the murine CatSper current, suggesting that, at least in mice, intracellular alkalinization is sufficient to open CatSper channels [46]. In addition to this, bovine [47] and equine [48] sperm are sensitive to intracellular alkalization, resulting in an influx of Ca^2+^ into the sperm. In fact, a histidine-rich domain exists at the intracellular NH2-terminus of CatSper and is related to the alkalization sensitivity of the channel [49]. However, a recent study found that EF-hand calcium-binding domain-containing protein 9 (EFCAB9), which was associated with CatSperζ, was of significance for the pH sensitive activation of CatSper [50]. In the current study, it was found that the proportions of sperm with high calcium ion concentration in the upper and lower layers of sperm were significantly increased when the sperm were incubated at pH 7.4 for 40 min, and that the increase in the upper layer was significantly higher than that in the lower layer. When NNC was added during incubation at pH 7.4, it was found that the calcium ion concentration, the proportion of forwardly motile sperm, and the proportion of X-chromosome-bearing sperm in the lower layer sperm decreased with the increase in the NNC concentration, and the correlation was extremely strong. Therefore, there is reason to believe that reduced motility of X-chromosome-bearing sperm is associated with low calcium concentration.

In effect, intracellular HCO_3_^−^ and Ca^2+^ bind to sAC, thereby increasing cAMP levels, which in turn activate PKA [51,52]. The binding of cAMP to the regulatory subunits of PKA allows dissociation of the tetramer and activation of the catalytic subunit. Once free, the catalytic subunits remain active to phosphorylate a wide variety of substrates on Ser/Thr residues [53]. PKA is responsible for the phosphorylation cascades that activate the different ion channels implicated in cytoplasm alkalinization [54], leading to increased membrane fluidity following cholesterol efflux, membrane hyperpolarization, and phosphorylation of several flagellar proteins, including axonemal and peri-axonemal proteins, and others involved in metabolism. PKA regulates many functional processes occurring in the germ cells: motility, capacitation, biochemical changes at the acrosomal and plasma membrane, the acrosome reaction, and fertilization [45]. Our results revealed that when sperm were incubated at pH 7.4, the sAC activities of the upper- and lower-layer spermatozoa were positively correlated with the cAMP/PKA activity and the proportion of forward motility in the corresponding layer. In addition, the pPKAs and pY contents of the upper layer spermatozoa were significantly higher than those of the control group. SQ22536 and H89 were added to pH 7.4 dilutions, respectively, and it was found that the protease activities of cAMP and PKA were significantly lower than those of the control group, the proportion of forwardly motile sperm in the upper layer of the test tube decreased, the proportion of X-chromosome-bearing sperm increased significantly, and the contents of pPKAs and pY decreased significantly. Although protein phosphorylation can promote the hyperactive motility of sperm, the maintenance of sperm motility requires large amounts of ATP, and there is a close relationship between sperm motility and energy metabolism. Studies have shown that various enzymes involved in glycolysis and the tricarboxylic acid (TCA) cycle enhance their activities through acetylation, thereby regulating the metabolic processes of glycolysis and the TCA cycle [55]. Our previous study found that the motility and the mitochondrial and glycolytic activities of the lower layer spermatozoa were lower than those of the upper layer spermatozoa after incubation at pH 6.2 and pH 7.4 (data not yet published). The ionic requirements and signaling pathways of mammalian sperm motility have been widely studied, but they are still not completely understood [56,57]. At the molecular level, sperm motility enhancement requires the activation of several signaling pathways, including but not limited to cAMP-dependent pathways, an increase in pHi, changes in [Ca^2+^]i concentration, and hyperpolarization of the sperm plasma membrane potential [33,58].

The sperm acrosome contains a large number of hydrolases, which are closely related to fertilization, and the integrity of the acrosome is conducive to the occurrence of fertilization [59]. The capacitation status of sperm can be assessed by monitoring calcium-mediated changes using the fluorescent antibiotic CTC. Neutral and uncomplexed CTC traverse the cell membrane of spermatozoa and enter intracellular compartments containing free calcium [60]. The CTC–calcium complex binds preferentially to hydrophobic regions, such as the cell membrane, resulting in different characteristic staining of non-capacitated, capacitated, and acrosome-reacted spermatozoa [61]. The proportion of intact and uncapacitated sperm in the upper and lower layers was not significantly different from the control group after incubation at pH 7.4 for 40 min. This is consistent with the results of previous studies that used enriched sperm for in vitro fertilization and found no difference in the fertilization rate between the test and the control groups [11]. A schematic diagram of the mechanism of the effect of the pH of the dilution solution on sperm motility is shown in Figure 10. 

## 4. Materials and Methods

### 4.1. Experimental Design

In this study, we conducted four main experiments. In experiment 1, we investigated the change in the pHi of dairy goat sperm when it was incubated in alkaline diluents. In experiment 2, flow cytometry and calcium colorimetric assay were used to detect the changes in [Ca^2+^]i concentration when sperm were incubated in an alkaline environment; after inhibiting the calcium ion channels, the sperm motility and the proportion of X-chromosome-bearing sperm in the upper and lower layers of the test tube were measured. In experiment 3, by sequentially inhibiting the protease activity of the sAC/cAMP/PKA pathway, detecting the motility activity of sperm in the upper and lower layers of the test tube, and determining the proportion of X-chromosome-bearing sperm, the signaling pathway that affected sperm motility was studied. In experiment 4, we used CTC staining to evaluate the capacitation status of the enriched sperm.

### 4.2. Ethics Statements

The Animal Ethics Committee of the Northwest A&F University approved the experimental procedures. All experimental procedures were conducted in accordance with the Guide for the Care and Use of Laboratory Animals (Ministry of Science and Technology of China, NWLA-2021-138).

### 4.3. Materials

Dairy goat semen was collected from purebred Guanzhong dairy goats raised at the China Cloning Animal Base in Yangling Demonstration Zone, Shaanxi Province. A pseudovaginal method was used to collect three samples of semen from each of 10 healthy rams aged 1.5–2 years (volume ≥ 1 mL, viability ≥ 70%), with a 48h interval between each collection.

### 4.4. Reagents

All chemicals and reagents were purchased from Sigma-Aldrich (St. Louis, MO, USA), unless otherwise indicated. Calcium channel inhibitor NNC 55-0396 dihydrochloride (HY-50722, MedChemExpress, NJ, USA), cAMP inhibitor 9-(tetrahydrofuran-2-yl)-9h-purin-6-amine (SQ22536, MedChemExpress, NJ, USA), PKA inhibitor H-89 dihydrochloride (HY-15979, MedChemExpress, NJ, USA), Anti-Phosphotyrosine Polyclonal Antibody ELISA Kit (abs120596, Absin, China), and Protein Kinase A Colorimetric Activity Kit (EIAPKA, ThermoFisher Scientific, MA, USA) were used in the experiments. Rabbit Anti-ADCY10 Polyclonal Antibody (bs-3916R, Bioss Antibodies, Beijing, China) was used in the immunoassays.

### 4.5. Sperm Diluent and Incubation Conditions

We referred to the basic non capacitative diluent formula of ram semen used by Muzzachi et al. [35] for the basic formula for the sperm diluent. In the current study, this was adjusted to glucose (55.51 mmol/L), fructose (55.51 mmol/L), lactose (29.21 mmol/L), ethylene diamine tetraacetic acid (EDTA) (20.82 mmol/L), CaCl_2_ (2 mmol/L), penicillin (100 U/mL), and streptomycin (100 U/mL). The osmotic pressure of the diluent was adjusted to 330 mOsmol/kg, and the diluent was prepared using a citric acid-sodium citrate buffer pair. The pH of the diluent in the control group was 6.8 (the pH of the semen in its natural state), and the pH of the diluent in the experimental group was adjusted to 7.4 using Tris.

Fresh semen samples (10 semen mixtures) were diluted to 1 × 10^7^/mL with pH 6.8 diluent and centrifuged at 1000× *g* for 5 min. The supernatant was discarded and 10 mL of different pH dilutions (n = 3) were slowly added. Test tubes were held at a 45° angle during incubation at 37 °C and 5% CO_2_.

### 4.6. Protein Extraction, SDS-PAGE, and Western Immunoblotting

Sperm (1 × 10^7^) incubated under different conditions were centrifuged at 16,000× *g* for 5 min, lysed with RIPA buffer containing complete protease inhibitors (Roche, Indianapolis, IN), heated to 100 °C, then held at 100 °C for 10 min. Extracted proteins (10 μg) were mixed with an equal volume of 2X SDS sample buffer, separated by SDS polyacrylamide gel (12.5%) electrophoresis, and transferred to PVDF membranes (GE Bioscience, Newark, NJ, USA). Western blots were incubated with the corresponding primary antibodies overnight at 4 °C (1:1000). After washing in TBST, enhanced chemiluminescence (ECL) detection was performed using an ECL system (GE Bioscience, Newark, NJ, USA), and the blots were exposed to film (Fujifilm, Tokyo, Japan). Band intensities were analyzed using Gel-Pro Analyzer (Media Cybernetics, Rockville, MD, USA).

### 4.7. TaqMan qPCR/PCR 

Using a DNAiso kit (9770A; TaKaRa) to extract genomic DNA from the frozen semen of Holstein dairy cows (Saikexing, Huhehaote, China) as a control [13], we extracted genomic DNA from the sperm of three male dairy goats to verify the existence of the sAC gene. Primers were designed according to the dairy goat sAC gene (Gene number, NM_018417.6) sequence in the GENE BANK database. The sequence was as follows: F-sAC:TGAAGTGGAGAAGACCTATTTGGA, R sAC: CCCACTCCTGGGTCTCAAAC. The PCR amplification products were subjected to TA cloning and transferred into DH5α *Escherichia coli*. The positive clones were selected for sequencing, and the results were analyzed by BLAST.

After incubation under different conditions, the upper and lower spermatozoa genomic DNA in the test tube were extracted, and the X-chromosome-bearing sperm count and ratio were calculated using the double TaqMan qPCR method (Bio-rad, Hercules, CA, USA) [11].

### 4.8. Detection of the Effect of Diluent pH on the pHi of Spermatozoa

Determination of pH changes inside and outside of sperm cells was conducted with reference to the experimental procedure of Achikanu et al. [62]. We previously established a standard curve for the change in internal pH of dairy goat sperm incubated in different pH dilutions, which was used directly in this test [11].

### 4.9. Determination of [Ca^2+^]i Levels

Sperm [Ca^2+^]i levels were assessed by flow cytometry using Fluo-4 AM. After incubation at different pHs, samples were centrifuged at 500× *g* for 5 min at room temperature. Fluo-4 AM (1 μM) was added to the samples, followed by incubation for 15 min at 37 °C in the dark. Data were recorded as individual cellular events using a BD FACS Canto II cytometer (BD, NJ, USA). 

The O-cresolphthalein complexone (OCPC) colorimetric detection of calcium ion concentration. The calcium ion standard was diluted with deionized water (n = 3). Detection working solution was added, and the samples were incubated in the dark at room temperature for 10 min. After incubation, the absorbance at 575 nm was measured with a microplate reader to establish a standard curve. Sperm were then collected and lysate was added for 5 min. This was followed by centrifugation at 10,000× *g* for 5 min. The supernatant was used to measure the absorbance and to calculate the calcium ion concentration. 

### 4.10. Sperm Motility Assessment by Computer-Assisted Sperm Analysis (CASA)

Aliquots (5 μL) of sperm suspension were placed into a Makler chamber that had been pre-warmed to 37 °C. CASA analysis was performed using a MaiLang digital image analyzer (SJ-TMDI100JZ, Nanning, China). For each sample, three independent cameras were used and a total of 24 random fields (eight fields/camera) were counted. The following parameters were measured: percentage of motile spermatozoa (total motility, %); percentage of progressively motile spermatozoa (progressive motility, %). Statistical comparisons were made using the unpaired Student’s *t*-tests.

### 4.11. Measurement of Proteasome Enzyme Activity

To determine whether the diluent pH affected the sperm motility in the sACY/cAMP/PKA pathway, motile sperm were incubated with 25 μM of SQ22536 [63] and 50 μM of H89 [13,21], respectively, for various times at 37 °C and 5% CO_2_. Sperm samples incubated under different conditions were resuspended in 1 mL of PBS. Then, each sample was split into three 1.5-mL tubes (500 µL each; 5 × 10^7^ sperm). For cell free measurements of adenylate cyclase activity, washed sperm were resuspended in lysis buffer and sonicated three times for 30 s each (JY88-IIN, Shanghai, China) with alternating 30 s periods of ice cooling. Assay conditions were as described by Jaiswal and Conti [64]. For the cAMP evaluation, a cAMP Direct Immunoassay Fluorometric Kit (ab138880, Abcam, Cambridge, United Kingdom) was used, following the manufacturer’s specifications. Fluorescence change was measured in a microplate reader set to top read mode at Ex/Em = 540/590 nm. For PKA evaluation, we used a PKA Kinase Activity Assay Kit (ab139435, Abcam, Cambridge, United Kingdom) according to the manufacturer’s instructions. Finally, 20 µL of stop solution was added to each well to stop the reaction, and the optical density at 450 nm was measured. For pY evaluation, we used the Anti-Phosphotyrosine Polyclonal Antibody ELISA Kit (abs120596, Absin) according to the manufacturer’s instructions. Detailed steps regarding the above kit operations are indicated in Appendix A.

### 4.12. Enriched Sperm Capacitation Status Assessment

The sperm capacitation status was assessed using the CTC fluorescence assay method, as described by Lee et al. [65]. CTC is a fluorescent antibiotic whose distribution in the spermatozoa changes during the transition from non-capacitated to capacitated and then to acrosome-reacted state, thereby allowing differentiation of the various steps of the sperm capacitation process. A total of 500 spermatozoa per slide were observed and different patterns of CTC-reactive spermatozoa were evaluated. 

### 4.13. Statistical Analysis

Statistical analysis was performed using GraphPad Prism 8.0 software (GraphPad Software Inc., San Diego, CA, USA). Data were expressed as means ± standard deviation of at least three independent experiments. Percentage data were transformed by arc-sin square root transformation to normalize the distributions before statistical analysis. Differences in variables between groups were evaluated by Student’s *t*-test (two-tailed). Multiple comparisons were performed using Duncan’s multiple range test, and the correlation between the two groups of data was detected by the Pearson product moment correlation coefficient using SPSS version 20 for Windows (SPSS Inc., Chicago, IL, USA) (*, *p* < 0.05; **, *p* < 0.01; ns, *p* > 0.05).

## 5. Conclusions

We found that when dairy goat sperm were incubated at pH 7.4, the pHi of the upper- and lower-layer spermatozoa increased to 6.9 at 10 min and then maintained a steady pH, while the calcium ion concentration increased with the incubation time. The percentages of total motility, forward motility, and [Ca^2+^]i concentration of the upper layer spermatozoa were significantly higher than those of the lower layer spermatozoa. The activity of sperm sAC protein was positively correlated with the concentration of calcium ions and alkaline dilution incubation reduces Ca^2+^ entry into X-sperm, and the motility was slowed down through the sAC/cAMP/PKA signaling pathway.

## Figures and Tables

**Figure 1 ijms-24-01771-f001:**
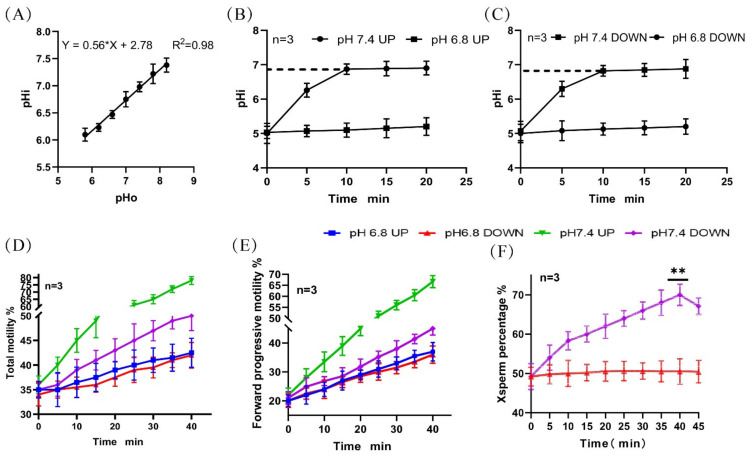
Changes in pHi and motility activity during sperm incubation. (**A**) Standard curve of the correlation between sperm pHi and diluent pH. (**B**,**C**) Changes in the pHi of upper- and lower-layer spermatozoa during incubation. (**D**,**E**) Changes in the proportion of total motile sperm and forwardly motile sperm in the upper and lower layers during incubation. (**F**) Changes in the proportion of lower-layer X-chromosome-bearing sperm during incubation. (**) indicates significant differences between the two groups (*p* < 0.01).

**Figure 2 ijms-24-01771-f002:**
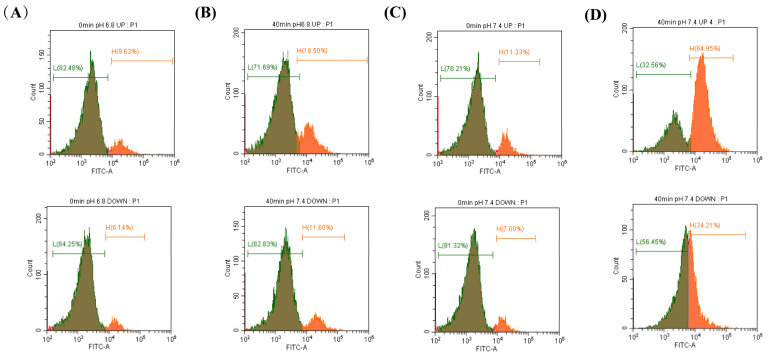
The changes in [Ca^2+^]i concentration during sperm incubation were detected by flow cytometry. (**A**–**D**) Proportion of sperm with high calcium ion concentrations in the upper and lower diluents when sperm were incubated at pH 6.8 and 7.4 for 0 min and 40 min.

**Figure 3 ijms-24-01771-f003:**
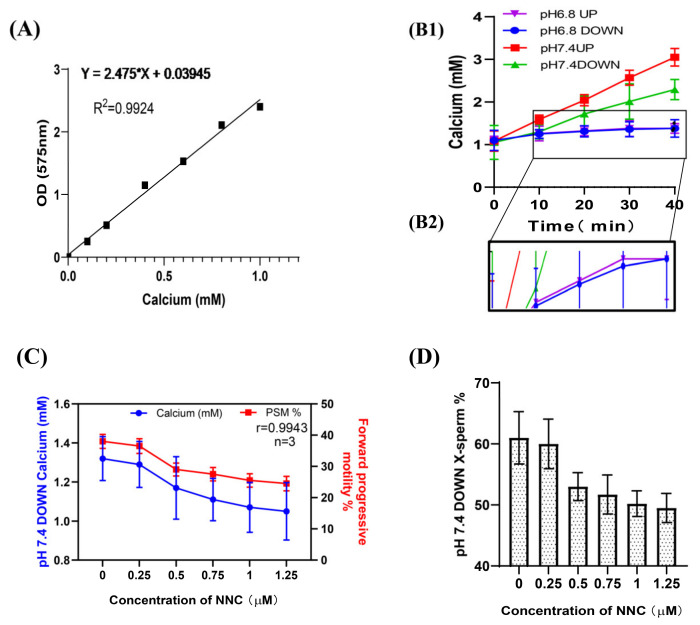
Changes in [Ca^2+^]i concentration during sperm incubation. (**A**) Standard curve of calcium ion concentration in sperm; absorbance at OD575. (**B1**) The [Ca^2+^]i in the sperm changed with the incubation time. (**B2**) A partially enlarged view of F1. (**C**) The calcium ion concentration of the lower-layer spermatozoa and the proportion of forwardly motile spermatozoa in the pH 7.4 dilution solution varied with the NCC concentration. (**D**) The proportion of X-chromosome-bearing sperm in the lower layer of the test tube in the pH 7.4 dilution varied with the concentration of NCC.

**Figure 4 ijms-24-01771-f004:**
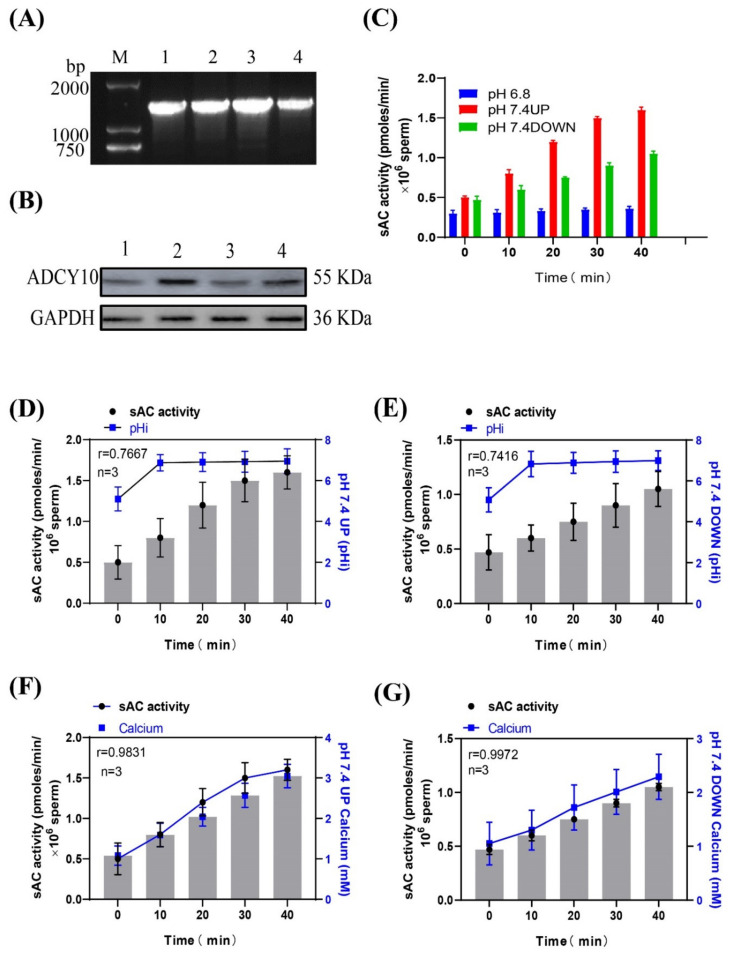
(**A**) Gel electrophoresis of the PCR amplification product of the sAC gene in dairy goat sperm. (**B**) Western blot of sAC from dairy goat sperm. (**C**) Changes in sAC protein activity when sperm were incubated in pH 7.4 dilution. (**D**,**E**) Relationship between sAC protein activity and pHi during sperm incubation. (**F**,**G**) Relationship between sAC protein activity and [Ca^2+^]i concentration during sperm incubation.

**Figure 5 ijms-24-01771-f005:**
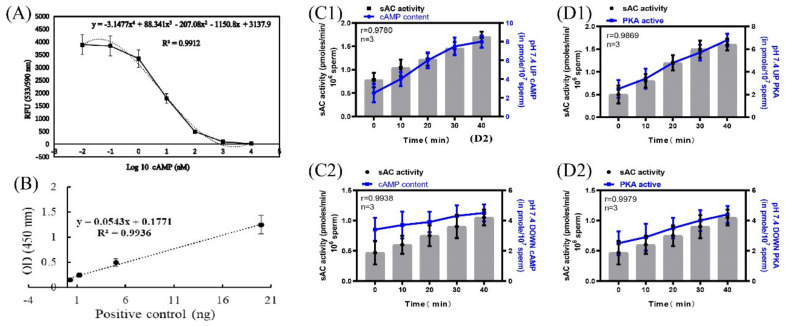
Correlation of sAC protein activity with cAMP and PKA protease activity. (**A**) Standard curve of cAMP content and OD533 RFU. (**B**) Standard curve of PKA concentration and OD450 RFU. (**C1**,**C2**) Correlation between sAC protein activity and cAMP content. (**D1**,**D2**) Correlation between sAC protein activity and PKA protease activity. Results are expressed as mean ± SEM, n = 3 experiments. Two-way ANOVA and Bonferroni’s multiple comparison test.

**Figure 6 ijms-24-01771-f006:**
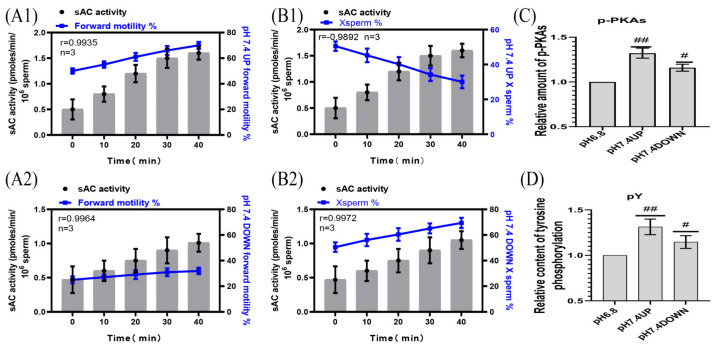
Correlation of sAC protein activity with sperm motility activity. (**A1**,**A2**) sAC protein activity was correlated with the proportion of forwardly motile spermatozoa. (**B1**,**B2**) Correlation between sAC protein activity and the proportion of X-chromosome-bearing sperm. (**C**,**D**) Statistics of the relative contents of p-PKAs and pY in dilutions of pH6.8 and pH7.4. Results are expressed as mean ± SEM, n = 3 experiments. Two-way ANOVA and Bonferroni’s multiple comparison test. (##) and (#) indicate significant differences between the two sets of data, respectively (*p* < 0.01) and (*p* < 0.05).

**Figure 7 ijms-24-01771-f007:**
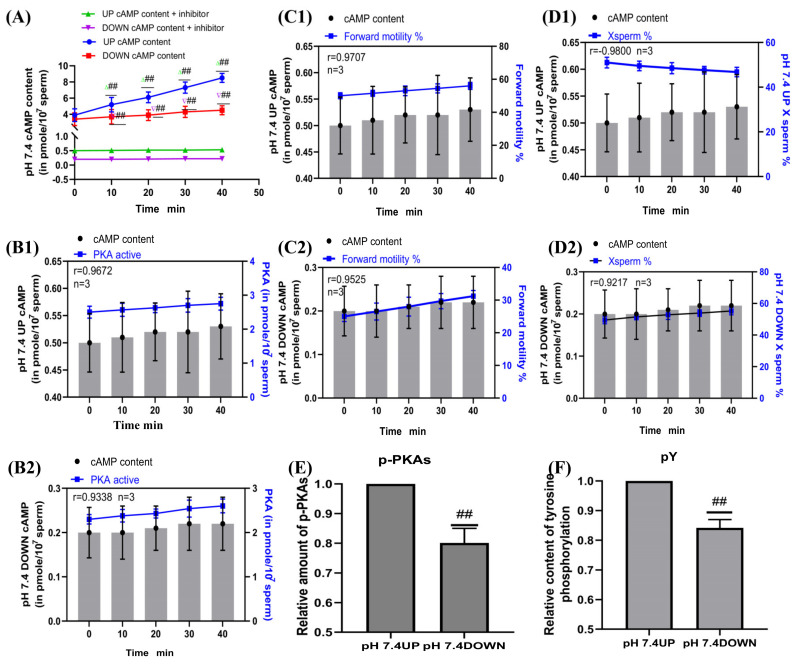
Sperm motility following inhibition of cAMP content. (**A**) Changes in cAMP content in sperm after SQ22536 supplementation. (**B1**,**B2**) Correlation between cAMP content and PKA protease activity of sperm. (**C1**,**C2**) cAMP content was correlated with the proportion of forwardly motile spermatozoa. (**D1**,**D2**) Correlation between cAMP content and the proportion of X-chromosome-bearing sperm. (**E**,**F**) Statistics of p-PKAs and pY content in sperm after the addition of cAMP inhibitors. Results are expressed as mean ± SEM, n = 3 experiments. Two-way ANOVA and Bonferroni’s multiple comparison test. (##) indicates significant differences between the two groups (*p* < 0.01).

**Figure 8 ijms-24-01771-f008:**
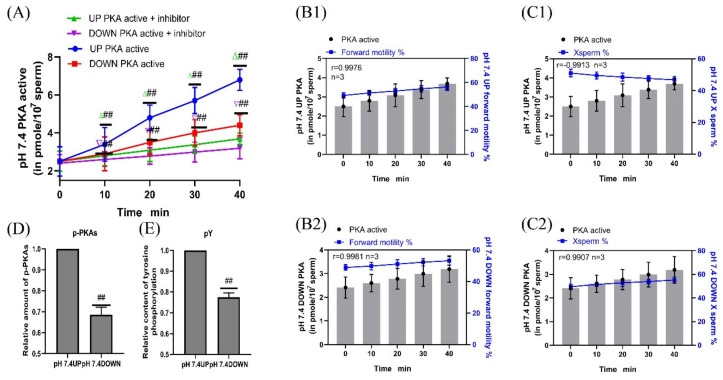
Sperm motility following inhibition of PKA protease activity. (**A**) Changes in PKA protease activity in sperm after H89 supplementation. (**B1**,**B2**) PKA protease activity was correlated with the proportion of forwardly motile spermatozoa. (**C1**,**C2**) Correlation between PKA protease activity and the proportion of X-chromosome-bearing sperm. (**D**,**E**) Statistics of p-PKAs and pY content in sperm after the addition of PKA kinase activity inhibitors. Results are expressed as mean ± SEM, n = 3 experiments. Two-way ANOVA and Bonferroni’s multiple comparison test. (##) indicates significant differences between the two groups (*p* < 0.01).

**Figure 9 ijms-24-01771-f009:**
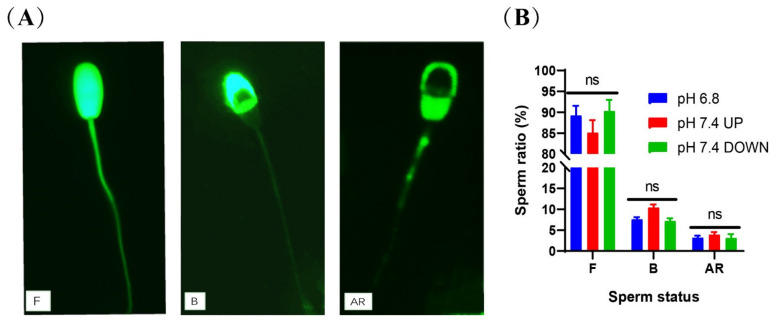
Detection of sperm capacitation status after incubation. (**A**) CTC staining method to evaluate sperm capacitation. F = live/non-capacitated sperm, B = live/capacitated sperm, and AR = acrosome-reacted sperm. (**B**) Statistics of capacitation status after sperm incubation. (ns) indicates no significant difference between the two groups (*p* > 0.05).

**Figure 10 ijms-24-01771-f010:**
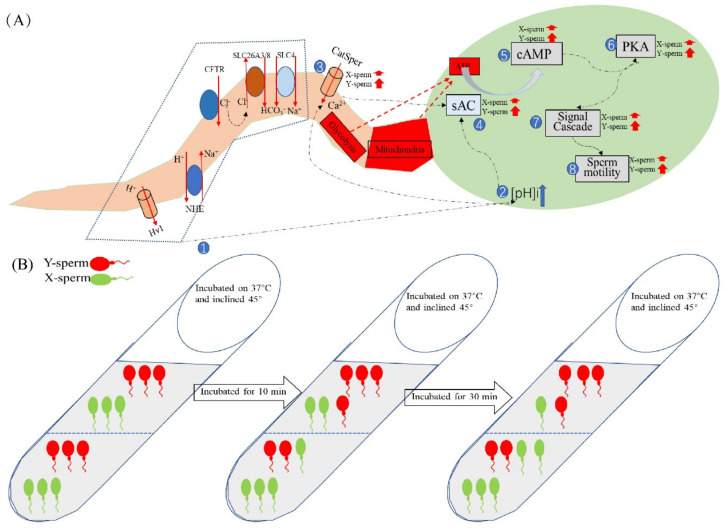
Schematic diagram of the mechanism by which diluent pH affects sperm motility. (**A**) Changes in pHi, [Ca^2+^]i concentration, and motility-related protease activity of X/Y-chromosome-bearing sperm during incubation in the alkaline diluent, with numbers indicating the order of changes within sperm. (**B**) Dairy goat sperm samples were incubated with pH7.4 dilution, the tubes were tilted at 45° and placed in a 37 °C incubator. The pHi of the sperm was about 5.0 at 0 min, and the ratio of X/Y-chromosome-bearing sperm in the upper and lower layers of the test tube was equal. The pH of the sperm increased to 6.9 after 10 min of incubation and remained unchanged thereafter, with a higher proportion of sperm in the lower layer than in the upper layer. When incubated for 40 min, the number of sperm in the lower layer and the proportion of X-chromosome-bearing sperm were significantly higher than those in the upper layer.

## Data Availability

All data, models, and code generated or used during the study appear in the submitted article.

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
