# Peer review of "Alkaline Dilution Alters Sperm Motility in Dairy Goat by Affecting sAC/cAMP/PKA Pathway Activity"

_ijms, 2023, doi:10.3390/ijms24021771_

Round 1

Reviewer 1 Report

In this study, 4 main experiments are performed. In experiment 1, the change in pHi of goat sperm when incubated in alkaline diluents is investigated. 

In experiment 2, using flow cytometry and calcium colorimetric assay, changes in [Ca2+]i concentration are sought when spermatozoa are incubated in an alkaline environment; and after inhibiting calcium ion channels, sperm motility and the proportion of spermatozoa X in the upper and lower layers of the test tube are measured. 

In experiment 3, by sequentially inhibiting the protease activity of the sAC/cAMP/PKA pathway, detecting sperm motility activity in the upper and lower layers of the test tube, and determining the proportion of sperm X, the signaling pathway affecting sperm motility was studied.

In experiment 4, the capacitation status of the enriched spermatozoa is evaluated by CTC staining.

It is a bit complicated to understand the article when the description of material and methods (section 4) is after the results (section 2) and the discussion (section 3).

The study is carried out with techniques appropriate to the results to be sought, adequately described. Likewise, the interpretation of the results is adequate and with an acceptable discussion.

 Some English terms and carefully review the formatting and word-break are need

(Fig. 5I).(Fig. 5H). were missing

(Fig. 5I).(Fig. 5H). were missing

Author Response

--1. Thank you very much for reviewing the manuscript. Your summary of the main findings of the study is complete.

--2. Thank you for your approval of this experimental design and manuscript writing.

--3. It is a bit complicated to understand the article when the description of material and methods (section 4) is after the results (section 2) and the discussion (section 3).

Response: As for the layout of the manuscript, we fully complied with the layout requirements of the International Journal of Molecular Sciences. We sincerely apologize for the trouble caused to you in the process of reviewing the manuscript.

--4. The study is carried out with techniques appropriate to the results to be sought, adequately described. Likewise, the interpretation of the results is adequate and with an acceptable discussion.

Response: Thank you for your affirmation of the logical relationship between section 4 and section 2 described in the manuscript, as well as your approval of the discussion section (section 3) in the manuscript.

--5. Some English terms and carefully review the formatting and word-break are need.

Response: Thank you for your suggestions on the writing of the terms in the manuscript, we carefully checked and revised the full text, and the revised parts are marked in red.

--6. (Fig. 5I).(Fig. 5H). were missing

Response: We sincerely apologize for the lack of pictures in the manuscript. We reorganized and analyzed the data in section 2.5 and supplemented and revised them in the revised manuscript.

Reviewer 2 Report

“Alkaline dilution alters sperm motility in dairy goat by affecting sAC/cAMP/PKA pathway activity” is an interesting and well written manuscript. The authors used so many techniques to explain the mechanism by which pH affects the motility of X sperm of goat. It is aimed to increase the female kid rate since it has benefits as both milk production and economical. The only suggestion that I can do is it may be better to define “X sperm” term as “X chromosome bearing sperm (X sperm) at the beginning of the manuscript.

Author Response

--1. Thank you very much for reviewing the manuscript.

--2. Thank you for your approval of this experimental design and manuscript writing.

--3. Thank you for your recognition of the content of this study and its significance for practical production.

--4. The only suggestion that I can do is it may be better to define “X sperm” term as “X chromosome bearing sperm (X sperm) at the beginning of the manuscript.

Response: Thank you for your suggestion, we have supplemented it in the revised manuscript, see line 33 for details.
